# Are Mechanical and Biological Techniques Efficient in Restoring Soil and Associated Biodiversity in a Brownfield Site?

**Guillaume Jacek** [1,2], **Anne Rozan** [1,3] **and Isabelle Combroux** [2,*]

1   École Nationale du Génie de l'Eau et de l'Environnement de Strasbourg (ENGEES), GESTE UMR MA 8101, F-67000 Strasbourg, France
2   Université de Strasbourg, CNRS UMR 7362 LIVE, Institut de Botanique, 28 rue Goethe, F-67000 Strasbourg, France
3   Université de Strasbourg, F-67000 Strasbourg, France
*   Correspondence: combroux@unistra.fr

**Abstract:** Brownfield technosol restoration represents a real opportunity to minimize land consumption, but it often requires intensive intervention and reconstruction, leading to soil donor site destruction and potential pollution. Using a former oil refinery brownfield site, this research compares the short-term (one year) impact of localized restoration methods, including soil surface scarification, organic matter addition, earthworm inoculation (locally and commercially sourced), on the soil fauna and vegetation properties. Vegetation rapidly covered the bare soil, but few significant impacts were observed in terms of the soil or vegetation structure. The initial positive impact of soil scarification on surface compaction quickly faded and earthworm inoculations did not seem to impact soil characteristics. Natural soil colonization by vegetation or fauna was the major driver in soil modification. The analysis of an intermediate reference showed a delay of multiple decades between vegetation, fauna density, and soil structure improvement, as well as the achievement of a soil of "good biological quality". We conclude that the use of engineer species on brownfield soil in its actual form is not an early restoration step but should be tested in later steps (e.g., when organic matter content is sufficient). However, additional research is needed in engineer species selection and multi-compartment approaches. A better understanding of the impacts of natural colonization in the early stages of technosol restoration is also essential for restoration techniques' improvement.

**Keywords:** soil health; earthworm inoculation; ecological restoration; wet meadow; soil scarification

## 1. Introduction

In addition to acting as a substrate for vegetation, soils play a key role in multiple processes, including biogeochemical cycling, biodiversity habitat provision, and extreme event mitigation [1–3]. They also contribute significantly to the global carbon cycle [1,4–8] and store approximately 2500 Gt of carbon [9]. However, current alterations in the ecological functions of soils (biological, hydric, climatic, and agronomic), resulting from their occupation or use, also known as artificialization, strongly affect their capacity to provide ecosystem services [10]. Therefore, soil loss-related concerns are increasingly an element of national [11,12], European [13,14], and worldwide [15,16] legislation and policies. Land-use policies are thus promoting soil recycling, restoration, and artificial space renaturation to compensate ongoing soil modifications.

In many countries, brownfield redevelopment, defined as the management, rehabilitation, and return to a beneficial use of a brownfield site [17], is seen as an opportunity for soil de-artificialization and the potential integration of ecological restoration measures [18–21]. However, the technosols of industrial sites are characterized by high levels of degradation, pollution, and artifact content in the surface layer [1,22–24]. The pedogenesis of these young soils is also limited by the low organic matter content and high compaction level. Their restoration, therefore, represents an important technical challenge with high restoration

thresholds. Conventional remediation methods often require invasive or ex situ treatment (e.g., biopiling, bioreactors, land farming), enhancing the level of biological and structural soil degradation [25,26] and overburdening their ecological restoration.

Soil restoration is a relatively new discipline [1,27,28], although many techniques already exist to restore various aspects of soil integrity. Mechanical methods, such as surface soil scarification, can reduce soil compaction, promote plant growth [29,30], and improve water infiltration [31–34]. Large-scale organic material additions from agricultural activities or green waste are used to increase nitrogen and carbon availability, thereby improving plant colonization and heightening microbial activity [35–39]. The transfer of entire soils is used to quickly rehabilitate the soil, enabling the transfer of both soil biotic and abiotic compounds in "whole-of-community" rewilding [40–43]. More recently, the use of engineer species, such as earthworms or ants, has been proposed to restore multiple aspects of degraded soils [4,44–46]. Earthworm activity improves various soil structural properties, including porosity (e.g., [47–49], aggregate stability, water infiltration, and the regulation of biogeochemical cycles [49–52]). All these methods could represent valuable opportunities to restore already artificialized soil, such as those found in brownfield technosols.

Despite their potential, the use of such ecological engineering soil restoration methods on brownfield sites remains limited. These methods could indeed be challenging to implement as they could present significant drawbacks linked to material sourcing (i.e., the partial destruction of a donor area) or to the unwanted introduction of species or chemicals during soil or green waste transfer [53–56] when too few precautions are taken. In addition, little is known about the impact on highly degraded soil of mechanic or small-scale methods, such as those used in localized scenarios and in controlled conditions (e.g., earthworm inoculation mesocosms [45]), leading to a knowledge gap in their transferability. Forey et al. [57] observed the positive impact of the earthworm species *Lumbricus terrestris* transfer on vegetation biomass in a former gravel site after only one year in semi-controlled conditions; however, the feasibility of earthworm transfer in on-field and highly degraded conditions still needs to be tested.

In this research, we tested the impact of soil restoration methods on various soil structures and functions applied in the field on severely degraded soils. We addressed the following questions: (1) Are physical or chemical restoration methods, such as mechanical approaches or the localized addition of organic matter, sufficient for restoring soil characteristics and promoting vegetation and fauna colonization in the early step of soil restoration? (2) Is commercial earthworm transfer tested during one year under semi-controlled in situ conditions in mesocosms by Forey et al. [57] efficient in real field conditions with highly degraded soils? (3) Does earthworm sourcing (commercially sourced vs. local species) have an impact on restoration success? Our study sites lie within a wet meadow created on a former industrial site as a compensatory land reclamation measure. The efficiency of the different restoration techniques was estimated by the study of above-surface vegetation, soil fauna communities, and soil physicochemical properties one year after treatment as Forey et al. [57] observed rapid changes in their experiment. The short time step of one year was chosen in order to monitor the early phase of brownfield evolution under the different tested techniques and the durability of the impacts of these techniques that could be impacted by the highly degraded aspect of the technosols.

## 2. Materials and Methods

### 2.1. Study Sites

#### 2.1.1. Experimental Site

The study was conducted on a restored wet meadow created on the former Reichstett oil refinery site. The site is located 15 km north of Strasbourg (France) (48°39′45.84″ N, 7°46′2.75″ E) and 5 km west of the Rhine River main channel (Figure 1). After 50 years of refining activities, the refinery closed in 2013, leaving behind a highly impacted industrial wasteland, heavily polluted with hydrocarbons. Soil remediation and dismantling

of infrastructure occurred between 2016 and 2018, and the site was redeveloped into a business park.

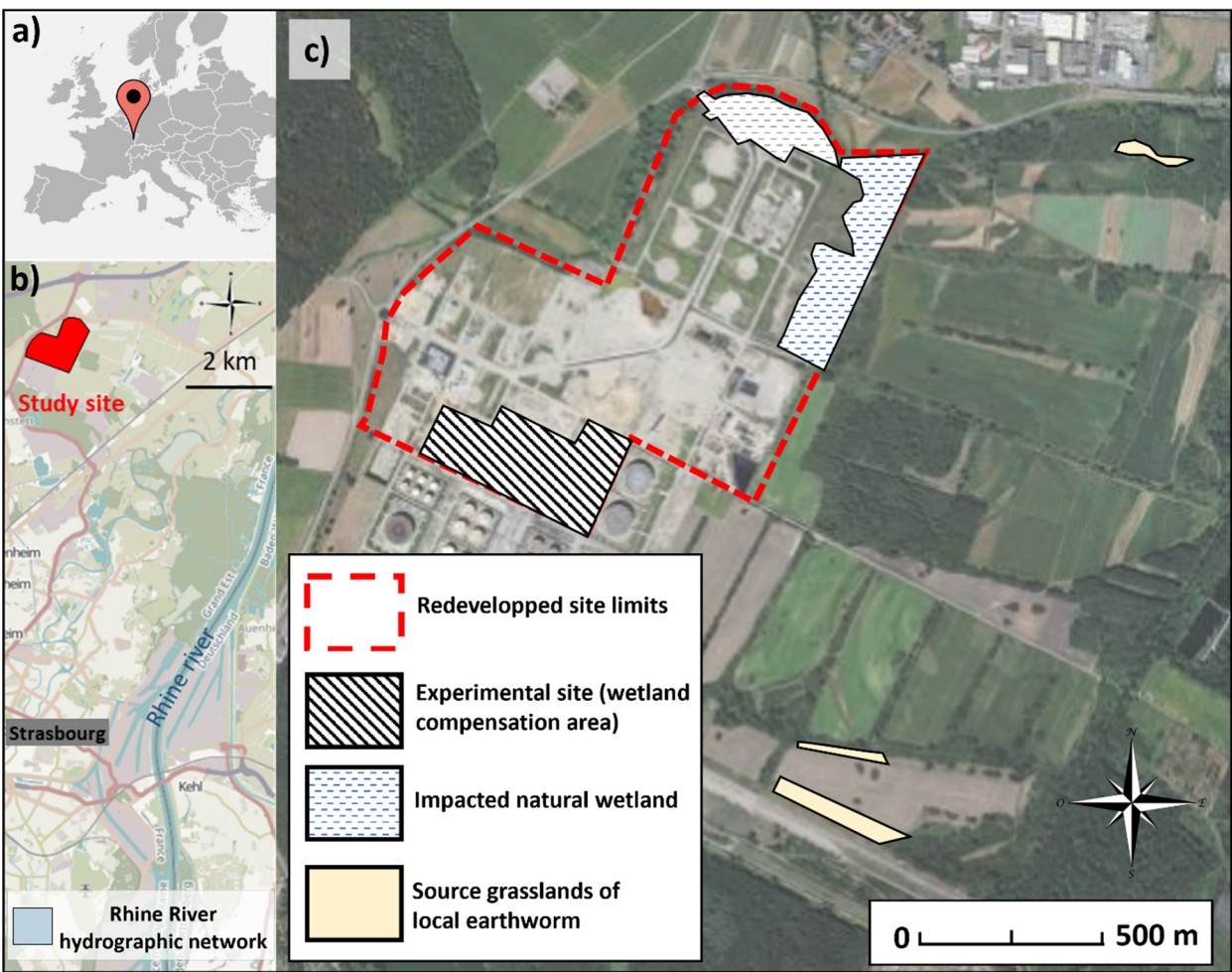

**Figure 1.** Study site location and delimitation. (**a**) Study site location in eastern France; (**b**) location of the site on the Rhine River floodplain; (**c**) experimental site and the grassland source area of the local earthworm communities. Map font sources: (**a**) Wikimedia common CC by SA2.0 [58]; (**b**) © OpenStreetMap, ODbl, CC-BY-SA 2.0 [59]; (**c**) Maps data, Google Earth,© 2022 Landsat/Copernicus [60].

Apart from the main brownfield land-use changes, the redevelopment also led to the destruction of a natural wetland, leading to the implementation of biodiversity offset measures. As the site is included in the Rhine alluvial floodplain, which is characterized by a high density of wet meadows and wetlands, this compensatory measure was directly implemented on the redevelopment area so that the experimental site is very close to the natural wetland destroyed by the project (Figure 1).

Ground excavation revealed a soil composed of sand (69%), silt (about 25%), and clay (6%), with an average pH of 7.9 and exceptionally low total carbon and nitrogen contents (0.47% and <0.05%, respectively). The restoration works brought the water table closer to the ground surface and created a wetland characterized by grasslands, distributed on a moisture gradient (the water table between 30 and 50 cm in depth on average) and two temporary ponds (Figure 2).

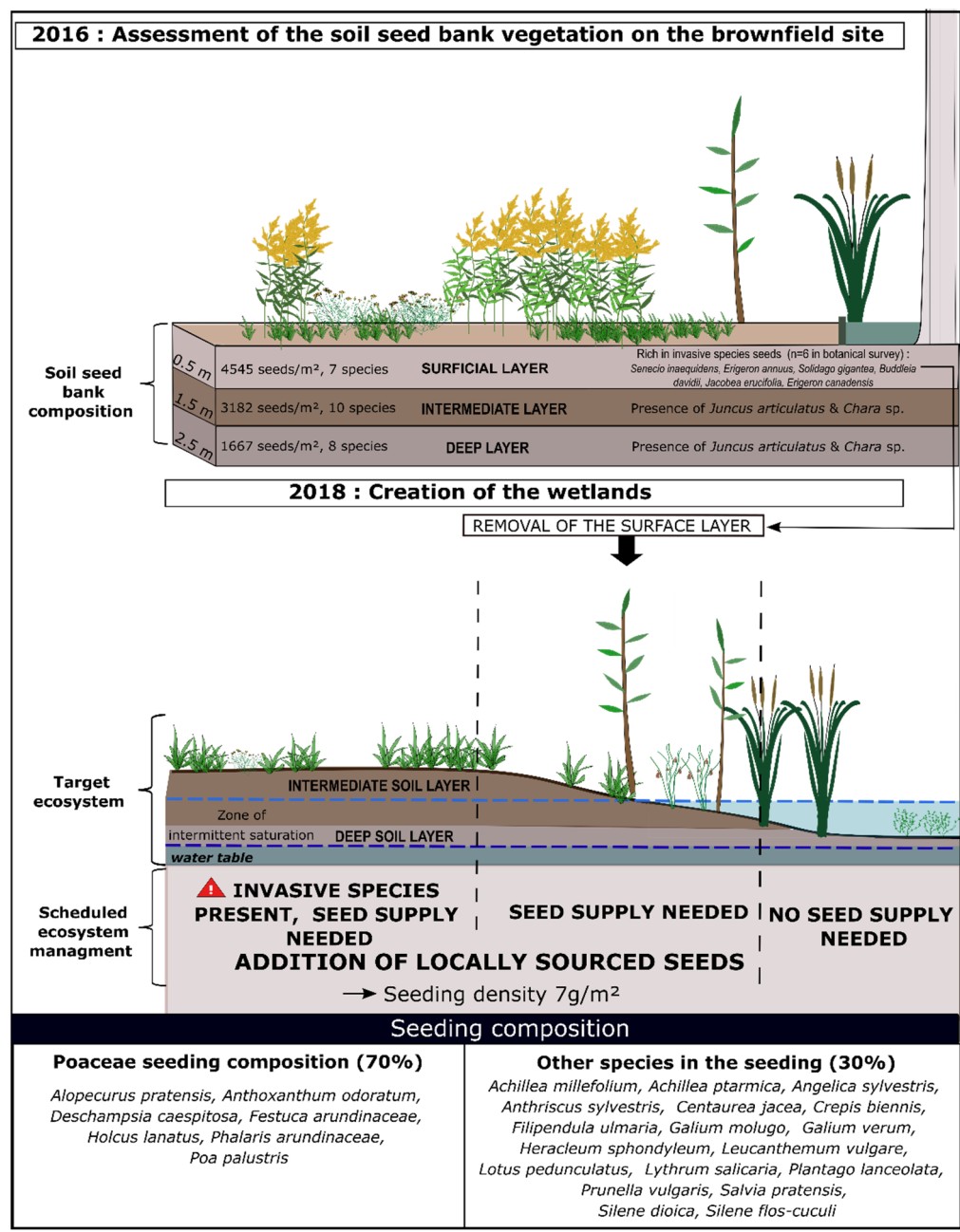

**Figure 2.** Vegetation restoration scheme for the created wet meadow habitats following an initial soil seed bank analysis and a botanical inventory of the brownfield site in 2016.

In 2016, a preliminary study (unpublished data) used the seedling emergence method to assess the site's soil seed bank [61,62]. This preliminary assessment highlighted a limited seed bank and concluded a high risk of plant invasion (Figure 2).

Therefore, the experimental site was seeded with a specific mix of locally sourced, 7 g/m$^2$, wet meadow species (Vegetal Local™) in May 2019. The seed mixture comprised 25 species, including 7 Poaceae (70% of the mix composition), 1 Fabaceae (*Lotus pedunculatus*), and a mix of 7 other species (Figure 2).

### 2.1.2. Reference Sites

We used two wet meadows as reference sites. Both sites are located on the Rhine floodplain. The Woerr site, located 50 km north of the experimental site (48°58′26.05″ N, 8°13′25.82″ E), constituted an intermediate reference (IRF) due the previous use of the soil

and the location. It is an alluvial grassland bordering a former gravel extraction area that closed in early 1990s. Therefore, it could have been impacted by heavy passage and storage related to gravel extraction and transportation equipment, and it was cultivated (extensive or intensive cultures depending of the years) until 2000 when it was restored (passive restoration) and managed by the Collectivité Européenne d'Alsace. This wet meadow has thus been a naturally functioning grassland for about 20 years. Its soil is characterized by an average concentration of silt (56%), sand (22%), and clay (22%) with a very low percentage of coarse elements (<1%).

The Schersand site, located 25 km south of our study site (48°27′5.34″ N, 7°43′51.16″ E), constituted a final reference site (REF). This wet meadow was managed by the Conservatoire des Espaces Naturels d'Alsace and has been a naturally functioning area for more than 50 years. Its soil is mainly sandy–silty (45% silt, 41% sand, and 14% clay with a percentage of coarse elements lower than 1%).

### 2.1.3. Earthworm Donor Sites

Three wet grasslands located in the agricultural areas 1 km east of the experimental site (Figure 1) were selected as earthworm donor sites. They were identified after an initial characterization of the earthworm density in the surrounding grasslands using the mustard extraction method [63]. Wet meadows presenting an earthworm density equal to or higher than 80 individuals per square meter were selected.

### 2.2. Soil Restoration Techniques

The experimental site is composed of an increasing gradient of soil humidity; to avoid drowning of earthworms, the area where methods were tested was limited to a part of the experimental site that was never flooded during the year. In order to allow a comparison between the different techniques, the area also had to be homogeneous at a larger scale, even if finer local variations, due to the heterogeneity of the technosols, could be present.

In addition to the dismantling of infrastructures and soil depollution; wet meadow species (Figure 2) were sown a few days before soil restoration techniques were applied to the plots in May 2019 (Figure 3): inoculation of a local multispecific earthworm community (EWL); inoculation of a monospecific community of commercially grown *Lumbricus terrestris* (EWC); soil surface scarification (SCA); and localized inoculation of organic matter (POS). We also delineated untreated control plots (CTL) where no additional restoration techniques occurred.

Restoration techniques were tested on 2 m² circular experimental plots and 5 replicates were completed per technique. Plots were randomly distributed on preselected area but with at least 7 m apart. This distance prevented, or at least limited, contamination by earthworms migrating from other plots during the year of testing, as Butt et al. [64] observed the maximum density of selected earthworms at an average of 10 m from the initial inoculation site after ten years. A total of 25 experimental plots were surveyed. These 25 plots were randomly distributed on the preselected area of the experimental site to limit the impact of local soil heterogeneity. All plots were seeded simultaneously a few days after treatment (May 2019) implementation, along with the rest of the wetland with the seed mixture shown in Figure 2.

For EWL, earthworms were collected in earthworm source grasslands (Figure 1) by hand without any sizing or sorting of species. For EWC, adults *Lumbricus terrestris* (from a fishing bait wholesaler), an anecic species already used in other soil restoration projects (e.g., [57]), were used. For EWL and EWC, inoculation units (EIUs) were composed of 3 L mixture composed of 60% organic potting soil, 35% presterilized compost, 5% coffee ground–limestone mixture (adapted from [65,66]), and 32 earthworm individuals (multiple species made up of approximately two-thirds anecic species and one-third endogenous species for EWL). Five EIU per plot were carefully inoculated at 30 cm to obtain a density of 80 earthworms/m² (Figure 3) and then covered with 5 cm topsoil.

This density is consistent with values found in a literature review (e.g., [57]) and the density observed in the surrounding grasslands. Special care was taken to limit soil disturbance during inoculation. Each earthworm plot was covered with a bird net for the first three months to reduce the impact of bird predation on the earthworm population. The EIUs, with total carbon and nitrogen concentrations of 30% and 1.2%, respectively, provided an initial supply of organic matter to facilitate the establishment of earthworms in the brownfield soil. For POS, we used a similar EIU mixture (but without any earthworms) and installation protocol as for EWL and EWC. Scarification (SCA) consisted of scraping the top layer of soil to a depth of 10 cm (Figure 3).

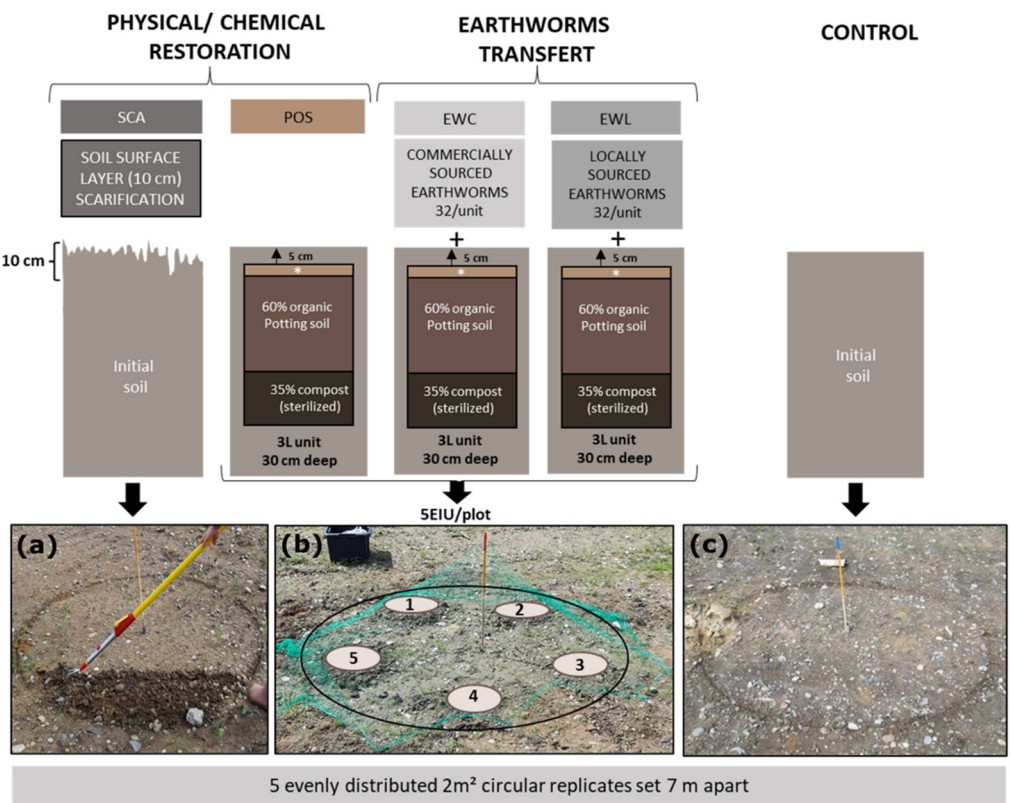

**Figure 3.** Description of the tested soil restoration techniques. Mechanical soil surface (10 cm deep) scarification (SCA, photo (**a**)). Earthworm inoculation techniques consist of the inoculation in the 2 m² plot of 5 earthworm inoculation units (EIU; pinked number 1–5 circles) technique each containing 32 commercially sourced earthworms (EWC), 32 locally sourced earthworms (EWL), or no earthworms (POS). Labels 1–5: equidistant location of the five EIU in the replicate protected from birds by a net (**b**). Control (CTL) corresponds to plot where no techniques were applied (**c**). Photo G. Jacek, 2019.

*2.3. Field Survey and Laboratory Analysis*

2.3.1. Initial (T0) and T + 1 Year Field Sampling Design

An initial characterization of the preselected area of the experimental site to identify baseline conditions was conducted from late March–early May 2019 and is hereafter referred to as T0. Soil destructive samplings (soil analysis and fauna sampling) were taken outside the experimental plots. Eleven sampling points were located in a systematic way in order to perform soil and fauna characterization of the experimental site. Nondestructive surveys (soil surface compaction, vegetation) were conducted in each of the 25 experimental plots.

Because of the COVID-19 lockdown, the T + 1 survey could not be conducted before late May 2020. As spring temperatures were high, the earthworms' maximum activity period had finished and the earthworm survey, e.g., through cast counting or soil sorting, could not be performed. Every other survey (destructive and nondestructive soil,

vegetation, and fauna samplings) on the experimental site was conducted on every experimental plot. The T + 1 survey was performed in the same period on the reference (IRF and REF) sites. Five 2 m$^2$ circular sampling plots were randomly distributed in each reference site. Soil, vegetation, and fauna samplings were performed on these plots using the same methods as in the experimental plots.

### 2.3.2. Soil Characteristics

At each of the 11 initial sampling points (T0) and experimental plots (T + 1), two topsoil samples of 520 cm$^3$ (S = 103 cm$^2$; H = 15 cm) were obtained using a corer for soil physico–chemical analysis. These samples were air-dried for four days during which big soil clods were manually broken off. One of the two samples was then weighed, and the 3–5 mm aggregates size class was sieved and used to assess the soil aggregate stability during slow and fast wetting using the Le Bisonnais [67] method (ISO 10930: 2012). This method evaluates the mean and weighted diameter of the aggregate after three water treatments. The second 520 cm$^3$ sample was sieved to 2 mm and dried again at 40 °C for 48 h. A sample of 20 g of dry sieved soil (<2 mm) was ground to a 250 μm thin powder and used to measure the total soil carbon and nitrogen contents using an elementary FLASH 2000 NC—Thermo Fisher Scientific™ analyzer with an accuracy of 0.05%.

A surface soil sample of 250 cm$^3$ (D = 8 cm, H = 5 cm) was also taken with a metal cylinder to assess the surface bulk density at each of the 11 sampling points (T0) and experimental plots (T + 1). Samples were dried in the oven at 105 °C for 24 h then weighed with a precision of 0.1 g. The bulk density was evaluated by estimating the ratio between the dry soil mass and soil sample volume (250 cm$^3$). Additionally, 10 random soil surface compaction values were measured at T0 and T + 1 on each experimental plot using a UTS-0078 pocket penetrometer, 0–5 kgf/cm$^2$.

### 2.3.3. Vegetation

Vegetation inventories were conducted at each plot at T + 1 (plots were unvegetated at T0). Plants were identified to the species level and their percentage cover was estimated. For the functional analysis, each species was assigned to one of the following functional groups: graminoids (including Poaceae, Juncacae, and Cyperaceae), leguminous species (Fabaceae), and other forbs (all other species). In each plot, a 30 cm × 30 cm sample of aboveground vegetation was cut at the plants' base (not including roots), sorted according to the functional groups, dried at 70 °C in an oven for three days, and then weighed to obtain the functional dry biomass, which was used as a surrogate for plant productivity.

### 2.3.4. Soil Fauna

At each of the 11 sampling points (T0) and experimental plots (T + 1), one topsoil sample of 520 cm$^3$ (S = 103 cm$^2$; H = 15 cm) was obtained using a corer. Soil fauna was extracted from these samples using the Berlese–Tullgren method [68]. Samples were exposed to a 70W halogen light for four days, creating a dry heat gradient in the soil sample previously placed in a funnel. A container at the end of the funnel containing 70% ethanol was used to recover and preserve the soil fauna. Invertebrate individuals were sorted and identified down to the class level. We divided the Collembola into three functional groups: epiedaphic, hemiedaphic, and euedaphic, following the classification of Bokhorst et al. [69].

### 2.4. Data Analysis

Statistical analyses were performed using R software [70]. We used univariate analyses to detect any significant effects of each of our treatments and significant differences with the initial conditions or the reference sites. We tested for normality and homoscedasticity using the Levene and Shapiro tests, respectively, with an α = 0.05; the results indicated whether subsequent analyses required a parametric or a nonparametric test. We ran one-factor ANOVA as a parametric test and Kruskal–Wallis for any nonparametric tests. To discriminate the effect of the various treatments on vegetation, we performed nonmetric

multidimensional scaling (NMDS) using the R package vegan [71]. NMDS condenses multidimensional information into a 2D representation [72]. Finally, we assessed the biological quality of the soils using the QBS (soil biological quality index) method [73,74], which is based on the analysis of the morphological characteristics of individual fauna. Standard deviation was used to represent the heterogeneity inside the sample. A Spearman test was used to assess the potential correlation between vegetation values, soil values, and soil fauna values of all the plots combined.

## 3. Results

### 3.1. Effects on Soil Characteristics

One year after the application of the additional restoration treatment, the soil physico–chemical characteristics exhibited few significant differences, as compared with T0 (Figure 4).

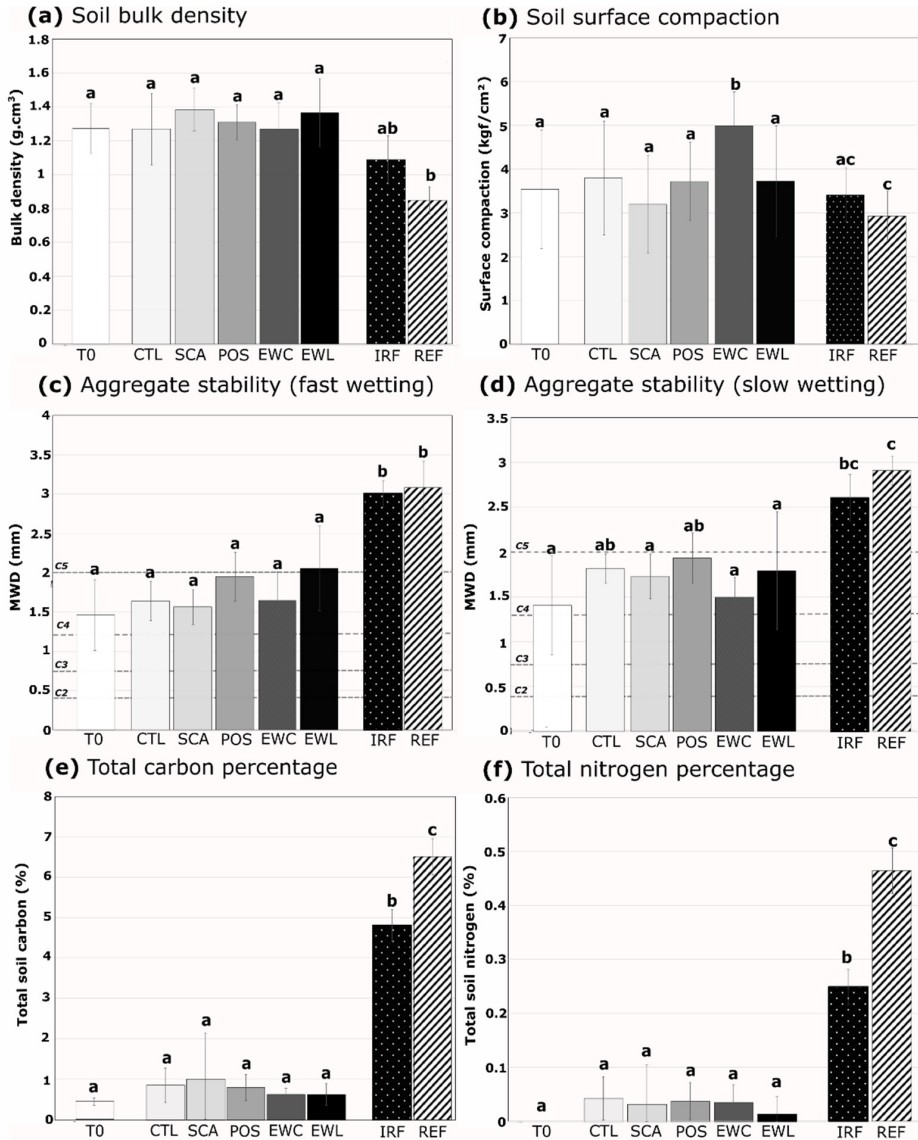

**Figure 4.** Soil characteristics (mean ± SD) of the sampling points at T0 and plots at T + 1 year. (**a**) Soil bulk density; (**b**) soil surface compaction; (**c,d**) stability of soil aggregates during (**c**) fast wetting and (**d**) slow wetting [67]. Mean weight diameter (MWD) corresponds to the post-treatment size of the lasting aggregates; classes of soil stability range from unstable (C2) to strongly stable (C4) (see [67]); (**e**) soil total carbon content; (**f**) soil total nitrogen content. Different letters above the bars (**a–c**) indicate significantly different values at a 95% level of significance according to the Kruskal–Wallis or ANOVA test. See the Section 2.2 for soil restoration technique abbreviations.

The exception was for soil compaction at the surface for commercial earthworm inoculation treatment, which returned a significantly higher value than the reference sites and exceeded 5 kgf/cm$^2$ (Figure 4b).

The values of soil bulk density, soil surface compaction, and aggregate stability after slow wetting tended to be similar to those reached by intermediate reference but were still significantly different from those reached in the final reference site (Figure 4a,b,d). The same pattern with a greater statistical significance was observed for soil organic carbon and nitrogen (Figure 4e,f).

*3.2. Effects of Soil Restoration Techniques on Vegetation*

At T + 1, all plots exhibited a high vegetation cover, with a mean total cover of 91%, and all plots ranged from 55% to full cover. No significant differences were highlighted between the various restoration techniques and control. The number of invasive species was slightly higher in the earthworm inoculation and scarified plots compared to control; however, we observed no differences in invasive species percentage cover (Table 1). The mean species richness of invasive plants ranged from 1.0 to 1.5 species per plot, predominantly being *Solidago gigantea* and *Senecio inaequidens*. Graminoids dry biomass was significantly lower in commercially sourced earthworm inoculation and organic matter addition, relative to the reference sites (IRF and REF). The reference plots (IRF and REF) had a significantly higher cover of graminoid species than the treatment plots. The cover of leguminous species was significantly higher (between 68.8% and 90.8%) on the experimental site than on the reference sites (13.7% and 2.7% for IRF and REF, respectively). REF was the only plot exhibiting significantly lower plant diversity, as it was dominated by *Bromopsis erecta* and exhibited a limited diversity of forbs and graminoids. A significant Spearman rank correlation coefficient was noticed between plant species richness and the total soil carbon content ($R^2 = 0.55$) and nitrogen content ($R^2 = 0.37$).

Differences in species composition were also observed between the experimental and reference sites (Figure 5a). NMDS clearly shows three distinct groups—at a stress value of 0.108—representing the final reference, intermediate reference, and the clustered soil restoration techniques (Figure 5a). *Vicia segetalis*, *Festuca* sp., *Dactylis glomerata*, and *Plantago lanceolata* species were found in similar proportions on all sites. *Lathyrus pratensis* and *Filipendula ulmaria* were only observed on the reference sites. Most of the leguminous species were found to be associated with the clustered soil restoration techniques. These results are in correlation with the higher cover of leguminous species in treatment plots.

The plant assemblages of the various restoration and control replicates were very similar (Figure 5b) and no differences were highlighted by the NMDS. Of the identified species, 35% were found in the initial seeding mix, and the other taxa were generally common pioneer species (Figure 5).

**Table 1.** Effect of restoration techniques on vegetation community characteristics (mean ± standard deviation, *n* = 5). Different letters indicate significant differences between treatments according to one-factor ANOVA tests and multiple comparison tests. See the Section 2 for soil restoration technique abbreviations. * $p \leq 0.05$, ** $p \leq 0.01$, *** $p \leq 0.001$, ns: no significant difference.

| Plant Communities | CTL | SCA | POS | EWC | EWL | IRF | REF | ANOVA Test between Techniques Only | ANOVA Test Including References |
|---|---|---|---|---|---|---|---|---|---|
| **Plant cover (%)** | | | | | | | | | |
| Total cover | 92.0± 17.9 [a] | 84.0 ± 18.8 | 96.0 ± 8.9 | 89.0± 11.4 | 92.03 ± 11.5 | 100.0 ± 0.0 | 99.4 ± 0.5 | ns | ns |
| Leguminous | 87.0 ± 19.9 [a] | 68.8 ± 26.1 [a] | 90.8 ± 19.7 [a] | 81.7 ± 9.4 [a] | 82.1 ± 20.7 [a] | 13.2 ± 19.1 [b] | 2.7 ± 1.3 [b] | ns | *** |
| Graminoids | 20.2 ± 14.4 [a] | 31.5 ± 19.7 [a] | 18.6 ± 24.8 [a] | 12.8 ± 10.7 [a] | 18.8 ± 22.8 [a] | 70.6 ± 15.4 [b] | 94.5 ± 6.7 [b] | ns | *** |
| Other forbs | 13.4 ± 4.1 | 23.5 ± 21.8 | 15.4 ± 6.3 | 13.2 ± 9.5 | 13.4 ± 8.3 | 37.0 ± 24.8 | 11.5 ± 12.8 | ns | ns |
| **Species richness (#m⁻²)** | | | | | | | | | |
| Total | 5.9 ± 1.6 | 6.7 ± 1.3 | 6.1 ± 1.4 | 5.9 ± 0.8 | 6.2 ± 1.4 | 5.8 ± 0.7 | 4.6 ± 1 | ns | ns |
| Leguminous | 1.9 ± 0.8 | 1.6 ± 0.4 | 1.8 ± 0.6 | 1.7 ± 0.7 | 1.8 ± 1 | 1.2 ± 0.5 | 0.9 ± 0.2 | ns | ns |
| Graminoids | 1.9 ± 1 | 1.8 ± 0.4 | 1.9 ± 0.6 | 1.7 ± 0.6 | 1.8 ± 0.9 | 1.4 ± 0.5 | 2.1 ± 0.4 | ns | ns |
| Other forbs | 2.1 ± 0.9 [a,b] | 3.3 ± 1.4 [a,b] | 2.4 ± 0.4 [a,b] | 2.5 ± 0.3 [a,b] | 2.6 ± 0.4 [a,b] | 3.4 ± 1.1 [b] | 1.6 ± 0.8 [a] | ns | * |
| **Diversity index** | | | | | | | | | |
| Total | 1.57 ± 0.3 [b] | 1.0 ± 0.49 [b] | 1.4 ± 0.1 [a,b] | 1.47 ± 0.4 [a,b] | 1.4 ± 0.2 [a,b] | 1.6 ± 0.3 [b] | 0.7 ± 0.3 [a] | ns | *** |
| Leguminous | 0.5 ± 0.3 | 0.86 ± 0.40 | 0.69 ± 0.2 | 0.6 ± 0.6 | 0.7 ± 0.2 | 0.5 ± 0.3 | 0.6 ± 0.3 | ns | ns |
| Graminoids | 0.8 ± 0.2 | 0.8 ± 0.31 | 0.9 ± 0.3 | 0.8 ± 0.3 | 0.8 ± 0.5 | 0.5 ± 0.5 | 0.3 ± 0.1 | ns | ns |
| Other forbs | 1.0 ± 0.6 [a,b] | 1.6 ± 0.32 [b] | 1.3 ± 0.2 [a,b] | 1.3 ± 0.2 [a,b] | 1.4 ± 0.2 [a,b] | 1.3 ± 0.1 [a,b] | 0.7 ± 0.6 [a] | ns | ** |
| **Dry biomass (g·m⁻²)** | | | | | | | | | |
| Total | 1203 ± 854 | 900 ± 643 | 935 ± 570 | 647 ± 373 | 878 ± 456 | 439 ± 100 | 333 ± 80 | ns | ns |
| Leguminous | 9150 ± 786 | 665 ± 545 | 812 ± 515 | 595 ± 37 | 729 ± 523 | 12± 11 | 3 ± 3 | ns | ns |
| Other forbs | 56 ± 69 | 34 ± 42 | 83 ± 73 | 37 ± 22 | 26 ± 31 | 24 ± 17 | 8 ± 13 | ns | ns |
| Graminoids | 231± 215 [a–c] | 201 ± 147 [a–c] | 41 ± 50 [c] | 15 ± 10 [c] | 124 ± 192 [a,c] | 402 ± 99 [b] | 322 ± 81 [a,b] | ns | *** |
| **Invasive species** | | | | | | | | | |
| Species (*n*) | 0.4 ± 0.5 [a] | 1.2 ± 0.4 [b] | 0.8 ± 0.4 [a,b] | 1.5 ± 0.5 [b] | 1.2 ± 0.4 [b] | 0.6 ± 0.5 [a] | 0 [a] | ns | *** |
| Cover (%) | 4.2 ± 4.0 | 2.0 ± 2.0 | 3.0 ± 5.0 | 7.2 ± 10.0 | 6.0 ± 8.0 | 1.0 ± 1.0 | 0 | ns | ns |

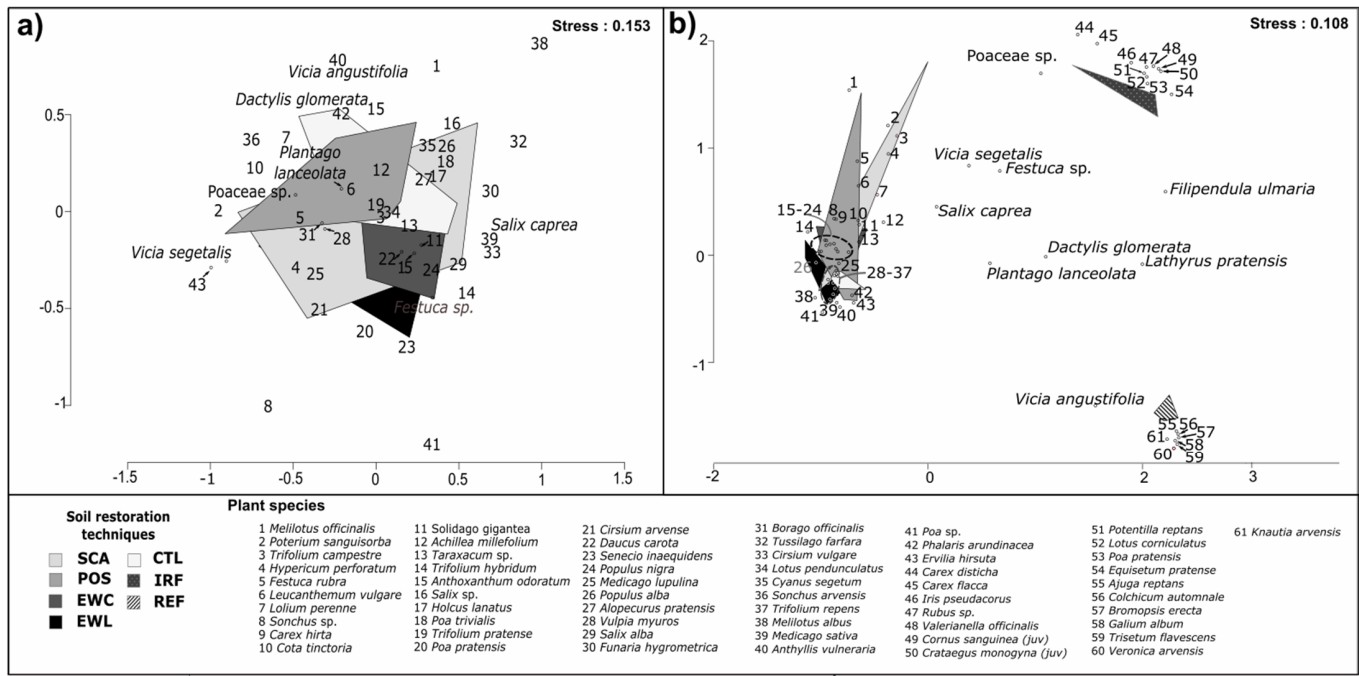

**Figure 5.** Nonmetric multidimensional scaling (NMDS) of plant species communities observed on the treated and reference sites. See the Section 2 for soil restoration technique abbreviations. (**a**) NMDS on experimental plots and references; (**b**) NMDS on experimental plots only to highlight the potential impact of the different techniques.

### 3.3. Soil Invertebrate Fauna and Soil Biological Quality

Soil Fauna Communities

At T + 1, the different additional restoration techniques presented no significant differences in soil fauna class richness, total individual density, or overall class composition, either compared to the reference sites or between the restoration techniques. In one year, however, class richness increased 6×, the diversity 4.5×, and individuals density 40× (Table 2), compared to T0. Plots inoculated with organic matter or commercially sourced earthworm were the only plots in which faunal diversity was not significantly different from T0 values. No significant differences between EWL plots inoculated with local earthworm community or organic matter and T0 were noticed for Acari density; scarified plots were the only ones with a statistically higher number of epidaphic collembolas than T0.

Soil quality based on faunal composition was significantly higher in all restoration plots than in T0 (Figure 6). We observed, however, no significant difference between additional restoration techniques (with the QBS value varying between 46.4 (EWC) and 62.2 (CTL)) or IRF (QBS = 50.4). They did not yet reach the soil quality value of REF, which was significantly greater (QBS = 98.8) and was the only site with a QBS value above that of the value for "good" soil quality calculated by Menta et al. [73], as a reference for a "good" soil.

Significant Spearman rank correlation coefficients were observed for the biological quality of the soil and the total soil carbon content ($R^2 = 0.39$) and nitrogen content ($R^2 = 0.4$—Supplementary Materials).

Similar levels of correlation were observed between the dry graminoid biomass and soil fauna class richness ($r^2$: 0.41) or the biomass of other species and hemi-density ($r^2 = 0.49$—Supplementary Materials).

**Table 2.** Effect of treatment on soil meso- and microfaunal communities (mean ± standard deviation, $n = 5$ and $n = 11$ for T0); see the Section 2 for soil restoration technique abbreviations. Statistically valuable difference was assessed using the Kruskal test, *** $p \leq 0.001$, ns: no significant difference. Statistical differences between treatments were tested using a pairwise test for multiple comparisons of mean rank sums (Dunn's test using Kruskal MC function in pgirmess r package). Different letters mean that samples are statistically different from each other.

| Soil Fauna Communities | T0 | CTL | SCA | POS | EWC | EWL | IRF | REF | Kruskal |
|---|---|---|---|---|---|---|---|---|---|
| Class richness | 0.8 ± 1.1 [a] | 6.0 ± 2.2 [b] | 5.2 ± 0.8 [b] | 3.6 ± 0.9 [b] | 4.2 ± 0.5 [a,b] | 5.8 ± 1.1 [b] | 5.2 ± 1.6 [b] | 9.0 ± 1.0 [b] | *** |
| Diversity index (Shannon) | 0.18 ± 0.33 [a] | 1.07 ± 0.31 [b] | 0.90 ± 0.27 [b] | 0.60 ± 0.32 [a,b] | 0.57 ± 0.23 [a,b] | 0.93 ± 0.32 [b] | 0.80 ± 0.22 [a,b] | 1.14 ± 0.26 [b] | *** |
| Individuals (ind·m$^{-2}$) | | | | | | | | | |
|     Total | 238 ± 410 [a] | 10243 ± 11252 [b] | 10088 ± 5525 [b] | 11388 ± 6739 [b] | 9176 ± 5445 [b] | 8012 ± 6681 [b] | 9215 ± 4807 [b] | 12086 ± 4886 [b] | *** |
|     Acari | 170 ± 340 [b] | 1300 ± 1249 [a] | 4171 ± 4662 [a] | 2212 ± 1560 [a,b] | 2794 ± 3981 [a] | 1106 ± 775 [a,b] | 6266 ± 2167 [a] | 5762 ± 4783 [a] | *** |
|     Collembola | 26 ± 50 [b] | 6984 ± 8692 [a] | 4171 ± 4503 [a] | 8866 ± 5770 [a] | 6111 ± 2899 [a] | 2929 ± 1688 [a] | 1998 ± 2700 [a] | 3356 ± 2875 [a] | *** |
|     Nematoda | 0 [b] | 873 ± 1794 [a,b] | 19 ± 43 [a,b] | 0 [ab] | 0 [a,b] | 39 ± 53 [a,b] | 349 ± 521 [a,b] | 621 ± 311 [a] | *** |
|     Coleoptera larvae | 15 ± 80 | 194 ± 168 | 58 ± 87 | 78 ± 126 | 19 ± 43 | 58 ± 126 | 78 ± 81 | 136 ± 111 | ns |
|     Diptera larvae | 15 ± 50 [a] | 504 ± 619 [a,b] | 1436 ± 1993 [b] | 175 ± 199 [a,b] | 58 ± 130 [a,b] | 3570 ± 6266 [a,b] | 175 ± 260 [a,b] | 233 ± 243 [a,b] | *** |
|     Other | 12 ± 50 [a] | 388 ± 434 [b] | 233 ± 130 [b] | 58 ± 53 [b] | 194 ± 119 [b] | 310 ± 269 [a,b] | 349 ± 340 [a,b] | 1979 ± 1893 [b] | *** |
| Collembola functional groups (ind·m$^{-2}$) | | | | | | | | | |
|     Epiedaphic | 15 ± 35 [b] | 369 ± 210 [a,b] | 834 ± 1169 [a] | 369 ± 269 [a,b] | 213 ± 372 [a,b] | 349 ± 311 [a,b] | 233 ± 130 [a,b] | 407 ± 520 [a,b] | *** |
|     Hemiedaphic | 9 ± 28 [b] | 5665 ± 8131 [a] | 3104 ± 3515 [a,b] | 7702 ± 2548 [a] | 3861 ± 1025 [a] | 2192 ± 1518 [a] | 1455 ± 2283 [a,b] | 2018 ± 2708 [a,b] | *** |
|     Euedaphic | 3 ± 17 [b] | 951 ± 1325 [a] | 233 ± 176 [a,b] | 795 ± 1008 [a] | 2037 ± 1669 [a] | 388 ± 369 [a,b] | 310 ± 419 [a,b] | 931 ± 1134 [a] | *** |

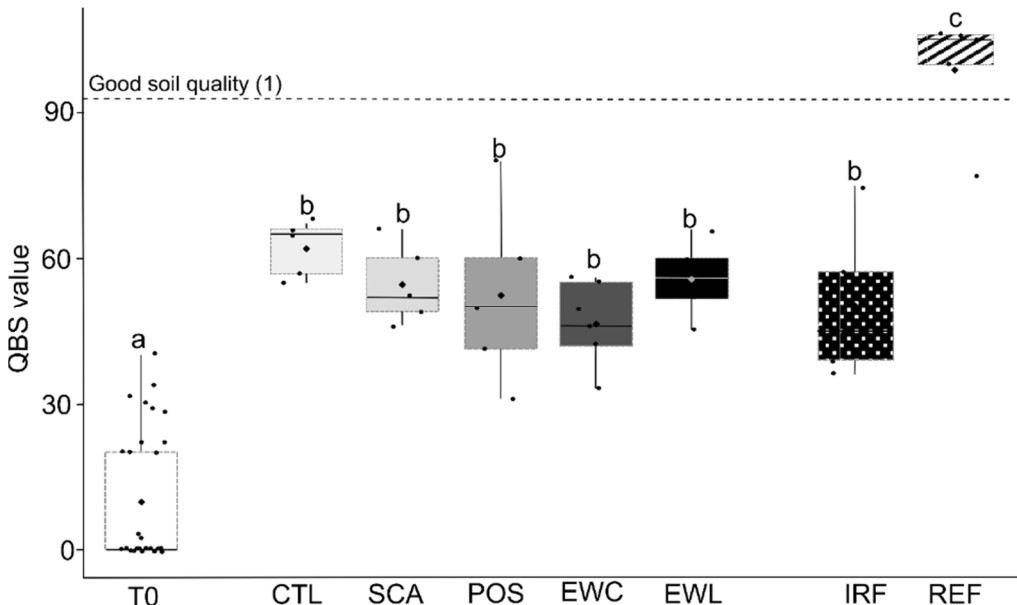

**Figure 6.** Boxplot of the soil biological quality results obtained using the soil biological quality index (QBS). Different letters above the bars (**a–c**) indicate significantly different values at a 95% level of significance by 1 factor ANOVA. See the Section 2 for soil restoration technique abbreviations.

## 4. Discussion

### 4.1. Effect of Abiotic Restoration Methods

Applying scarification methods or organic matter deposits in addition to classical depollution and sowing restoration methods did not enhance or accelerate ecosystem structure and function recovery after one year.

Edwards et al. [75] observed that scarification of the soil surface followed by seed dispersal promoted a greater plant diversity. By uncompacting and loosening the soil surface (Figure 3b), soil scarification creates new opportunities for seeds to germinate [76,77] and facilitates the penetration of seedling roots into the soil. This easy seed germination may have temporarily favored the germination of some invasive species arriving from the surrounding environments, potentially explaining the greater invasive species richness observed on the scarified plots, compared to the control. In general, the Rhine and alluvial floodplains constitute efficient corridors for invasive populations [78,79] so that restoration methods can create new opportunities for seeds to germinate, especially in relatively adverse abiotic conditions, such as soils with a poor nutrient content (Figure 4) or of poor quality (Figure 6), which could lead to exotic plant invasion [56]. However, it is important to take into account that only two invasive species have been inventoried and that no positive impact on the cover of invasive species has been observed. The absence of other differences in plant composition among the tested soil restoration techniques suggests that the seed source (sowing, seed bank, and natural seed rain) is a primary factor to consider when attempting to ensure the success of soil restoration projects.

The fact that scarification did not have a significant impact on soil surface compaction after one year confirms that the loosening effect of soil scarification appears to be temporary and fades quickly, as was observed by Busso and Perez and, even after one year, by Drescher et al. [32,80].

The absence of any effect of organic matter inoculation on the soil chemistry could relate to the selected inoculation methods. We chose to inoculate the organic matter as a Kernel batch and not as a thin homogenized layer to mirror the earthworm inoculation units. Since the soil fauna had not completely recovered its function (Figure 6), the added organic matter was not fully mobilized by the soil fauna, and thus did not affect the total plot area. This organic matter was also hardly detected in random soil sampling, which explains the relatively low carbon content, even for POS (Figure 4). The correlation analysis,

however, confirms that the vegetation, soil fauna, and soil biological quality are dependent, at least in part, on the carbon content of the soil.

### 4.2. Effect of Earthworm Inoculation

Because of the COVID-19 lockdown in spring 2020, earthworm survival and population density could not be assessed and no individuals or trace were found in the sample used for soil structure or fauna analyses. The relatively few significant effects of inoculation with this engineer species, therefore, could not be assigned to the potential low survival rate of the individuals. However, a low availability of organic matter (Figure 4e,f) or a very high bulk density (Figure 4a) in the experimental plots may have favored a greater earthworm death or migration to more suitable soils. The survival of earthworms in chemically or physically degraded soils is a major question in most inoculation projects and field experiments [81]. Earthworms quickly seek out more suitable areas after site degradation [82,83]. Degraded soils generally tend to be avoided by earthworms, thus leading to a lower density of earthworms in these sites [84].

In this field study, we did not observe any earthworm inoculation effect on either plant biomass production or diversity one year after inoculation, contrary to the work of Curry and Boyle and Forey et al. [57,85] who tested earthworm inoculation on brownfield sites in enclosed mesocosms. Very few impacts related to inoculation on plant or fauna species composition were observed, despite other studies showing that earthworm communities play a key role in the composition of grassland plant communities [86].

In the literature, the majority of the positive effects related to earthworm inoculation on the physical properties of the soil were recorded in micro- or mesocosms. Very few field-based studies have been produced. In their review, Jouquet et al. [45] only listed three papers published between 1999 and 2014 showing positive results related to earthworm in-field inoculation. Our results suggest that mesocosm results cannot be directly transferred to the field. Inoculation density has to be adapted or soil abiotic conditions have to be modified to offer an ecological niche that is suitable for earthworm survival. Furthermore, such restoration techniques have to be tested in a way in which earthworm survival and dispersal can be surveyed, as Forey et al. [57] recorded an 85% mortality rate in their *Lumbricus terrestris* community one year after inoculation.

Finally, our results suggest the existence of an effect of earthworm sources on some restoration efficiency parameters, since commercially sourcing communities promoted more soil surface compaction than control and other techniques, including local population. This is in line with the work of Contos et al. [43] in which they strongly recommend the active use and consideration of the transfer of local invertebrates communities. These communities may contain environmentally adapted species, the presence of which increases the likelihood of successful implementation and ensures the establishment of a community with the appropriate functional and life history traits [43]. Other factors, such as the time needed for restoration [87,88], the choice of species [89], and the survival rate, can also explain our results.

### 4.3. Colonization Processes of Restored Soil

Despite a lower seed density in the seeded mixture, leguminous species cover was higher in the first successional stage of the experimental site than in the older grasslands of the reference sites. This highlights the crucial role played by nitrogen-fixing species in the colonization of extreme substrates, since they improve various soil attributes, such as organic matter content, soil structure and porosity, fertility, microbial structure, and composition [90]. In accordance with the course of natural succession in grasslands, graminoid species were more developed in older reference sites.

It has been shown that the composition of the vegetation, at a specific level, can influence the soil fauna, especially due to the interspecific interaction between the plant and the soil fauna [91]. Therefore, such changes in plant dominant species during the colonization process, e.g., the increase in biomass of grass species could have a positive

impact on the diversity of soil fauna classes by promoting a greater diversity of potential interspecific interactions. Our results highlighted a correlation between graminoid biomass and soil fauna class richness.

We observed rapid colonization of the soil fauna at the experimental sites in terms of both the number of individuals and classes with the composition dominated by acari and collembolas in all the treatments and references. This rapid colonization of newly created soil is consistent with other soil restoration or creation projects that also observed a rapid growth in soil fauna density [57,92–95]. A density averaging around 10,000 indviduals/m$^2$ correspond to the colonisation rate observed on other technosols or brownfield sites after one year [57,96]. The fact that, in our study, we did not observe any significant impact of the different treatments on soil fauna composition, contrary to what Forey et al. [57] observed, shows that allowing natural colonization could be sufficient to observe a rapid soil colonization.

However, even if the soil invertebrate fauna quickly reached a biodiversity level similar to the reference site, good soil quality is not reflected by soil fauna after only one year. Vanhé and Devigne [97], who studied collembolas community colonization on coal mine spoil tips, highlighted that collembola community composition strongly differs to the surrounding environment due to substrate composition and vegetation cover. A dominance of colonizing and pioneer species was observed on bare and low carbon content soils, such as coal mine spoil tips. This colonization pattern, with dominant pioneer species during the first years is also valid for other taxa that the soil is composed of [95,96]. The dominance of pioneer species less dependent of soil quality could explain the results observed with the QBS indicators.

These results, considering both vegetation and soil fauna, highlight the importance of natural colonization in the early stages of brownfields soil restoration. They also highlight the need for a better understanding of the importance of the early colonization processes on the success of restoration. The designing of soil restoration methods should take into account the potential of natural colonization. As an example, the use of engineer species could be a viable approach for severely degraded soils, providing inoculation is not done at T0 but when organic matter content is sufficient due to vegetation being at an early stage of colonization.

Brownfield soil is often highly degraded and reaching the reference state through active restoration could be complex and expensive. Passive restoration, as proposed by Prach et al. [98], could be an effective approach to limit the cost of restoration and produce fair results in terms of soil function and ecological dynamics. Yet, passive restoration often comes with slow recovery and potentially unwanted ecosystem properties [98]. Therefore, the advantage of passive versus high active restoration, such as soil transfer in the case of brownfield and highly degraded soil, still needs to be studied.

### 4.4. Soil Biologic Quality Restoration as a Long Process

Although natural colonization seems to be sufficient in our study to observe significant changes in the first year of restoration, achieving good ecological soil quality is a long-term process. Without judging the initial soil quality, after more than twenty years of restoration, the intermediate site continued to have a similar biological soil quality to that of the one-year post-treatment soils. The experimental site also exhibited rapid colonization by plants and soil fauna, although attaining the optimal conditions in terms of species composition remains a long-term process. Soil parameters must be fully restored through the activity of plants and must engineer fauna species and their complex relationships in order to reach a good soil quality, as shown by the correlation found. Soil pore size (related to the degree of soil compaction) and the availability of soil organic matter, for example, can directly affect the composition of the mesofaunal community [99–101]. However, soil biological quality also depends on diverse species' colonization processes. For example, after afforestation, Dunger and Voigtländer [96] observed that former mine sites were rapidly colonized by a large number of pioneer species of Collembola and Acaris. However, about ten years

were needed for colonization by slower colonizing species, such as earthworms, which then led to strong competition and a drastic decrease in the density of individuals. A slower and more progressive colonization of these restored sites was then observed.

The inoculation of engineer species, such as earthworms, produces promising results in a controlled condition. However, the constraints related to the supply of earthworms and their potentially low survival rate make it complex and costly to apply in situ to large areas. The introduction of earthworms alone in its current form does not appear to be the optimal method for restoring brownfield soils. In parallel, the impact of scarification, used in the decompaction of large areas, although initially effective in terms of the decompaction of the surface layer of the soil, seems to have very temporary results, with an essential reversion to the initial conditions being observed only one year after the experiment. These less invasive techniques seem to be far less efficient after their first years than techniques that modify soil compartment as whole, such as the addition of a massive amount of green waste compost to manufactured soils [102]. However, further studies are still needed to assess the long-term effects of such techniques and the differential impact of natural colonization from surrounding environments versus active restoration techniques.

**Supplementary Materials:** The following supporting information can be downloaded at: https://www.mdpi.com/article/10.3390/land11122133/s1. Table S1: Table of the correlation measured with the Spearman method between all the variable measures in the 3 compartments (soil structure, aboveground vegetation and soil fauna). The cells highlighted in fluorescent yellow correspond to the main observed correlations usually with an $r^2$ greater than 0.4 (except for carbon/QBS correlation).

**Author Contributions:** G.J.: conceptualization, methodology, formal analysis, investigation, writing—original draft, and visualization. A.R.: conceptualization, validation, and writing—review and editing. I.C.: conceptualization, methodology, validation, and writing—review and editing. All authors have read and agreed to the published version of the manuscript.

**Funding:** This research is part of the SEFIRR project, supported by the French agency for ecological transition (ADEME) (Grant No. 1972C0009) and the BF2 Rhein Park and of the Woerr project supported by the Department du Bas-Rhin.

**Acknowledgments:** The authors thank all the field and laboratory assistants, particularly Francesca Celentano and Etienne Chanez, for their help in the field and their advice in setting up the experiments and Martine Trautman for her help in soil analysis. We give special thanks to the Communauté Européenne d'Alsace and the Conservatoire des sites Alsaciens (CSA) for site access.

**Conflicts of Interest:** The authors declare no conflict of interest.

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
