# Peer review of "Are Mechanical and Biological Techniques Efficient in Restoring Soil and Associated Biodiversity in a Brownfield Site?"

_land, doi:10.3390/land11122133_

Round 1
Reviewer 1 Report
This paper on the effects of plot scale restoration treatments of a former industrial sites is of global significance due to the need to restore these sites.
This is a reasonably replicated field study that also provides useful data on an intermediately restored site and a reference site.
As noted in my numerous comments on the attached PDF file, the authors need to carefully refine their statements of results. There are a number of statements that are inconsistent with non-statistically significant results.
There are too many speculative discussion points with little or no convincing evidence from the results. This is particularly true in regards to the earthworm additions in which monitoring did not occur soon enough due to Covid restrictions.
The dominant result is 'no signficiant differences between treatments and the control plots'. This is actually 'good news'. In the case of this study, natural colinisation appears to be sufficient. Expensive interventions like seeding, scarification, organic matter additions and earthworm additions may not be needed at this early stage of restoration. Time (decades) alone may be sufficient to achieve reference conditions. This point needs to be clearly made, rather than inferred.

Author Response
Dear Editor and Reviewers,
Please find attached a corrected version of our manuscript based on both reviewers and editor comments. This revised version also take into account the English editing of the text. We thank reviewers and editor for their comments and advice that allowed us to clarify and complete our analyses.
Problems highlighted by reviewers in the study design explanation were corrected in order to clarify the implementation of the methods in the field and mainly to clarify the random aspect of the placement of the experimental plots. We have acknowledged the opinion of the reviewers that some proposals in the discussion are too speculative in relation to the statistical significance of our results; we have therefore modified part of the discussion to fit more closely to our analyses (see detailed aswers below)
We hope that our corrections and additions will answer their questions.
Reply to reviewer 1 :
“As noted in my numerous comments on the attached PDF file, the authors need to carefully refine their statements of results. There are a number of statements that are inconsistent with non-statistically significant results.”
Thank you for your comment, we have changed/modified some of the formulation in our results to better match the statistical reality of some of our results and to be more specific about which treatments (including the baseline) were statistically different from each other. See the following lines for examples:
L.325-332
L.368-373
L.379-381
Others comments on the attached PDF file :
Abstract and text have been modified to encounter reviewer 1 remarks. Some more specific answers are listed below:
Introduction
- 36 : desartificialization which is not commonly used in English have been replaced by ‘artificial space renaturation’ (now l. 42-43)
l.37 Brownfield redevelopment definition, and other recommendation have been implemented (now l. 44-45)
Material and method
“Not clear if these plots were all seeded with the mixture shown in Fig 2. Was the seeding done at the same time as these treatments?”
We clarified that seeding happened after the implementation of the treatment and at the same time for all the plots See lines 179-181
In the results and discussion part, the use of abbreviations to name methods has also been reduced
Discussion
“again, key result - 'do nothing' (allowing natural colonization) appears effective at end of first year”
Thank you for this pertinent remark which we have developed extensively in the improvement of the discussion line 518-532 and 588-590. See also the abstract.
“no evidence that this is ecologically significant”
Changes have been made in the part where our analysis seems to have been too speculative. See the entire discussion (line 284-400). The abstract has also been changed to better emphasize the lack of impact of the various techniques and the importance of natural colonization.
“There are too many speculative discussion points with little or no convincing evidence from the results. This is particularly true in regards to the earthworm additions in which monitoring did not occur soon enough due to Covid restrictions.”
As you pointed out, we may have been too speculative about the analysis of some of our results, especially since we have few that appear statistically significant. We modified some parts of our discussion to make them more appropriate to the results we had. We emphasized some points such as natural colonization, differences between baselines and the initial state where the statistical differences were clearer. We also deleted some parts. The discussion part was overall improved in this way and you can see the changes that were made on lines 468-471, 466; 452-454; 586-589. See also lines 547-556 and 471-474;413-414 which have been deleted but still appear in the correction tracker
“The dominant result is ‘no signficiant differences between treatments and the control plots’. This is actually ‘good news’. In the case of this study, natural colinisation appears to be sufficient. Expensive interventions like seeding, scarification, organic matter additions and earthworm additions may not be needed at this early stage of restoration. Time (decades) alone may be sufficient to achieve reference conditions. This point needs to be clearly made, rather than inferred.”
Indeed, this part was not sufficiently developed in our previous version despite its importance to explain the results we observed. We have added a reflection on natural colonization in our summary but especially in the discussion. We also discuss the benefits associated with potential passive or partially passive restoration of this soil type. See lines 518-532 and 588-590
Author Response
Dear Editor and Reviewers,
Please find attached a corrected version of our manuscript based on both reviewers and editor comments. This revised version also take into account the English editing of the text. We thank reviewers and editor for their comments and advice that allowed us to clarify and complete our analyses.
Problems highlighted by reviewers in the study design explanation were corrected in order to clarify the implementation of the methods in the field and mainly to clarify the random aspect of the placement of the experimental plots. We have acknowledged the opinion of the reviewers that some proposals in the discussion are too speculative in relation to the statistical significance of our results; we have therefore modified part of the discussion to fit more closely to our analyses (see detailed aswers below)
We hope that our corrections and additions will answer their questions.
Reviewer 2:
“As fully noted by the authors, these were 1-year results which limit the utility, but still could be valuable information to add to the literature. The authors should justify the use of 1-year data.”
Thank you for your advice. We have completed the explanation of the use of one year's data at the end of the introduction and in the discussion.
One year data were used to:
- Allow the comparison of in situ mesocosm condition used in Forey et al (2018) who found significative impact of earthworm inoculation after only one year, to the direct in situ inoculation we’ve done in our experimentation that could represent harsher conditions for inoculated earthworm
- Better understand early steps of brownfield soil evolution and colonization depending on the different technics and non-action
- Assess short term perennity of the tested methods on brownfield soil.
We hope that these changes better highlight the relevance of using one-year data. See lines 90-97
“2. The study design appears to be an issue. Unless I misunderstand the field layout (Fig 3), the treatments were not applied randomly in the area or in blocks. Each treatment was in the same limited area. Five reps were within a larger circle, where each rep was a specified distance from each other. There did not seem to be any attempt to address spatial variability. A couple of explanations within the methods section would help here: “
Thank you for this comment which seems to highlight that some flaws were present in the explanation of our study design. Indeed, 5 different treatments were applied. Each of the treatments was applied in 2m² plots and 5 replicates of 2m² were made per treatment, for a total of 25 plots (5 plots inoculated with local earthworms, 5 plots inoculated with commercial earthworms, 5 scarified plots, 5 plots with added organic matter and 5 control plots where no action was taken).
Our experimental site presents an increasing moisture gradient with some parts being under water for some time of the year. To avoid drowning of earthworms, a non-flooded sub-area was selected to implement the experiment. In this area, the 25 plots were randomly placed to avoid the impact of local soil heterogeneity. However, all plots were separated by at least 7m to limit earthworm contamination from one plot to another.
We have modified the explanation of the study design to make it clearer and to emphasize that randomization was done and justifies the validity of the statistical tests used: see lines 159-181
“Why was this approach chosen as it ignores the effect of spatial variability in brownfield sites, which randomization or blocking are meant to address? “
We used a total randomization of the experimental plan.
The random placement of each of our treatments and their associated replicates allows us to avoid potential biases related to the localized heterogeneity of brownfield soils. We hope that the transformation we made in our study design (lines 159-181) better highlights this randomization and justifies the validity of the test we used.
“How does this study design violate assumptions of the statistical tests employed? At times the field situation may require such a setup. If so, it would be useful for the authors to justify the study design. “
Non-random placement of the experimental plots with all replicates in the same area would have been potentially problematic and would not have guaranteed the independence of the measurements made on the different replicates. The local conditions related to the heterogeneous soil typical of brownfield technosols, as you rightly pointed out, would have prevented a proper comparison of the different techniques. However, this was avoided during the implementation of the experiment by randomly placing the 25 plots throughout the experimental area. The conditions of the statistical tests were thus respected here. The description of the field protocol has been clarified to make this aspect clear. See lines 159-181
“Do the authors believe that the lack of 1-year results could be affected by the inability to account for spatial variability in the brownfield site, given that such variability is generally large for these situations? “
Variability in brownfields is often important due to the impact of previous activities. As you suggested, local variation could have impacted some of our results due to local variability. To avoid those problems, as shown by the limited standard deviation of T0 in Figure 4 when describing the initial soil characteristics (compaction, carbon/nitrogen content, and aggregate stability), we first chose a globally homogeneous area of our experimental site to implement our entire experiment. To avoid further impact of finer soil variability, the plots were placed randomly. While this does not mean that results in one specific plot couldn’t be impacted by specifics conditions, the study design address this issue and limits the impact of soil heterogeneity. Therefore, we do not believe that heterogeneity explains the lack of results but rather the initial harsh soil conditions. See study design description line 159-181
“If I am wrong about the field design, it would be useful to better describe the setup. “
The setup has been now more precisely described, we hope it is now easily understandable line 159-181
“Figure 4. Bulk density I assume is given in units of g/cm3. This should be given on the axis or in the figure title. “
The unit has been added
Results and Discussion:
This study showed very few significant effects of any treatment, so we are looking at a paper based on null-results. While this can be useful in the literature, the questions on study design hold greater significance. “
We hope that the change in our description of the study design was helpful. We also highlight in the discussion the results in terms of natural colonization, which was the main driver of change in our study.
“The authors provide an honest interpretation of the tables and figures with just a few exceptions:
Lines 403-410: It is stated that the more local variety of earthworms was more effective at reducing soil strength. What the data shows is that the local variety produced a result that was no different than the control. What happened is the commercial variety appeared to increase soil strength. That is a distinction that would be useful to make. “
Thank you for this relevant remark. We did modify this part and showed that in fact the inoculation of commercially sourced earthworms seems to increase soil surface compaction. See line 307-309.and 468-470
In the Discussion, when comparing results, from published reports to these data, it will be most useful to note, for the papers cited, how many years after treatment the authors were reporting. Was it also first year results? If not, then further clarification may be useful.
The fading of measurable impact of scarification on soil surface compaction after only one year has also been observed in other studies. This has been clarified in the discussion. (see line 424 to 427). The impacts measured by Forey et al. (2018) after inoculation with L. terrestris were also measured one year after inoculation. We note that other publications analyzing the colonization of technosols by soil fauna have obtained the same results after 1 year (see lines 502-504).